# Classic and New Markers in Diagnostics and Classification of Breast Cancer

**DOI:** 10.3390/cancers14215444

**Published:** 2022-11-05

**Authors:** Roman Beňačka, Daniela Szabóová, Zuzana Guľašová, Zdenka Hertelyová, Jozef Radoňák

**Affiliations:** 1Department of Pathophysiology, Faculty of Medicine, P.J. Šafarik University, 040 11 Košice, Slovakia; 2Center of Clinical and Preclinical Research MEDIPARK, P.J. Šafarik University, 040 11 Košice, Slovakia; 31st Department of Surgery, Faculty of Medicine, UNLP and P.J. Šafarik University, 040 11 Košice, Slovakia

**Keywords:** breast cancer, histological and molecular subtypes, molecular biomarkers, breast cancer gene candidates, new generation sequencing, microRNA

## Abstract

**Simple Summary:**

With ever-increasing incidence, breast cancer is considered a most diagnosed type of cancer among women worldwide. Breast cancer arises through malignant transformation of ductal or lobular cells in female (or male) breast and the genetic, phenotypic and morphological heterogeneity has an effect on tumour’s behaviour, thereby instigating a need for individual personalized therapy. A traditional assessment of tumour’s characteristics involves a biopsy and histological analysis of a tumour tissue, and in recent years has been accompanied by analysis of molecular biomarkers to enhance the results. In this work we aimed to thoroughly investigate the latest data in this field of study and give a comprehensive review of novel molecular biomarkers of breast cancer and methodologies used to analyse them.

**Abstract:**

Breast cancer remains the most frequently diagnosed form of female’s cancer, and in recent years it has become the most common cause of cancer death in women worldwide. Like many other tumours, breast cancer is a histologically and biologically heterogeneous disease. In recent years, considerable progress has been made in diagnosis, subtyping, and complex treatment of breast cancer with the aim of providing best suited tumour-specific personalized therapy. Traditional methods for breast cancer diagnosis include mammography, MRI, biopsy and histological analysis of tumour tissue in order to determine classical markers such as estrogen and progesterone receptors (ER, PR), cytokeratins (CK5/6, CK14, C19), proliferation index (Ki67) and human epidermal growth factor type 2 receptor (HER2). In recent years, these methods have been supplemented by modern molecular methodologies such as next-generation sequencing, microRNA, in situ hybridization, and RT-qPCR to identify novel molecular biomarkers. MicroRNAs (miR-10b, miR-125b, miR145, miR-21, miR-155, mir-30, let-7, miR-25-3p), altered DNA methylation and mutations of specific genes (p16, BRCA1, RASSF1A, APC, GSTP1), circular RNA (hsa_circ_0072309, hsa_circRNA_0001785), circulating DNA and tumour cells, altered levels of specific proteins (apolipoprotein C-I), lipids, gene polymorphisms or nanoparticle enhanced imaging, all these are promising diagnostic and prognostic tools to disclose any specific features from the multifaceted nature of breast cancer to prepare best suited individualized therapy.

## 1. Introduction

According to GLOBOCAN 2020, female breast cancer (BC) is the most diagnosed malignancy worldwide (~48 per 100,000 inhabitants) and also the leading cause of cancer death (~14 per 100.000 inhabitants), surpassing the long-time first position of lung cancer [1,2,3]. In combined statistics, irrespective of sex, BC incidence is the highest among all estimated body cancers together with lung cancer (each share 11.6% of all 36 cancer sites) [1]. BC affects women as well as males [3]. Eurostat data show that standardized death rate from BC in 2016 was 32.7 and 0.6 per 100,000 inhabitants for women and males, respectively. In persons aged 65 years and over rates reach 67.2 per 100,000 inhabitants. BC accounted for ~7% of all deaths from female cancer. Mortality from BC in high-income countries is steadily decreasing, meanwhile in low- and middle-income countries it is increasing, and half of new cases are diagnosed in an advanced cancer stage [4,5,6,7]. In the latest Eurostat report, Slovakia was among countries with the highest BC incidence and cancer death rate, which certainly requires substantial attention [8,9,10].

BC is etiologically, histopathologically and genetically a heterogeneous disease with both hereditary predispositions and non-hereditary factors [2,6,11]. This is certainly true for BC as it refers to mammary carcinoma from ductal or lobular cells in the mammary epithelial tissue. Only a minor portion are sarcomas transformed from connective tissue and vessels [12]. Malignant transformation in BC is the product of accumulations of consecutive mutations in critical regions of the genome that are normally involved in control of cell growth and division, DNA repair and programmed cell death [13]. These mutations are partly inherited but mostly spontaneous. Contribution of genetic factors in BC has been indicated by familial occurrence which is estimated as 5–10% of all cases. High-penetrance genes which are linked with inherited BC susceptibility include *BRCA1* and *BRCA2,* and more rarely *TP53*, *PTEN*, *CDH1* and *SKT11* [14]. Heterozygotic mutations in DNA repair genes *BRCA1* (locus Ch17q21.31) or *BRCA2* (Ch13q13.1) are the most common inherited conditions associated with BC. Absolute risk of BC for BRCA1 mutations reads ~50–65% in females and ~1% in males, while in BRCA2 mutations female risk ranges from 40% to 55% and reaches up to 9% in males [3,15]. Familiar susceptibility to BC is also associated with mutations of lower penetrance genes as *ATM* (Ch11q22.3), *PALB2* (Ch16p12.2) and *CHEK2* (Ch22q12.1). Mutation in the androgen receptor gene (AR) has been found in cases of male BC [16]. Susceptibility to sporadic BC cases can be linked with many more genes: e.g., sporadic invasive ductal variant of BC and lobular BC are associated with somatic mutation of genes *RAD54L* (Ch1p34.1) and *CDH1* (Ch16q22.1), respectively. Other genes candidates associated with sporadic BC include: *TP53* (Ch17p13.1), *SLC22A1* (Ch11p15.4), *PIK3CA* (Ch3q26.32), *ESR1* (Ch6q25.1-q25.2), *RB1CC1* (Ch8q11.23), *KRAS *(12p12.1), *AKT1* (14q32.33), *RB1* (Ch13q14.2), *PPM1D* (Ch17q23.2), *MYC* (Ch8q24.21)*, FGFR1* and eventually *ERBB2* (Ch17q12), *CCND1* (Ch11q13.3), *GATA3* (Ch10p14)*, MAP3K1* (Ch5q11.2) in certain lineages [16,17,18,19,20,21].

Acquired factors may increase the incidence of BC, although the roles of many remain elusive [22]. Non-modifiable and modifiable risk factors of developing BC include age (>55 years), earlier onset of menarche, later menopause, dense breast tissue (more glandular and fibrous tissue) and benign breast conditions (squamous and apocrine metaplasias, fibrosis, adenosis, mastitis or fat necrosis) [17]. The rates are higher also for women whose first pregnancy was after age 30, who did not breastfeed or never had children, women who were on hormonal replacement therapy, obese, with low physical activity and unhealthy living habits such as cigarette smoking and alcohol consumption [17,22,23].

Despite the recent progress in identification of molecular diversity of BC and efforts made in a treatment, incidence and mortality rates of BC are still very high and challenging. New biomarkers are being tested for the diagnostic purposes and to estimate a prognosis and relapse of the disease [18,23]. The advantage of molecular diagnostic methods is the relatively easy acquisition of a sample for analysis, which is moderately inexpensive and provides fast results [4,6]. Such methods include the use of next-generation sequencing (NGS), RT-qPCR, in situ hybridization, microRNA tracking and multigene assays [24]. The purpose is to provide the best suited personalized therapy in order to avoid of overtreatment and side effects in patients with good prognosis, yet to provide sufficient therapeutic effect to the patients with worse prognosis [25]. It is also important to consider an economic relevance of methods to be designated as standard care [4,6,18,20].

## 2. Diagnostics and Surveillance of the BC

Most of current breast-cancer-related diagnostic procedures fall into one or more of the following categories: 

**(a) *Screening tests*** are done routinely to people who are not suspected of having BC. Self-performed manual palpation of breast (BSE) guided by professionals is considered an effective way of early detection of breast tumour. Findings of lumps, redness, thickenings or asymmetries in breasts and enlargement of axillary lymph nodes should be examined by medical professionals as the obvious next step. Regular screening has been one of the major tools in mortality decline over the past decade [6,7,21,26]; 

**(b) *Diagnostic tests*** are done in those suspected of having BC, either because of symptoms or screening results. These tests include mammography [27], or MRI [28], or biopsy, or a combination of all in uncertain cases [29]: Mammography and breast NMRI (Magnetic Resonance Imaging) are useful non-invasive ways of how to exclude eventual other palpable breast lumps as abscess, cysts or fibroadenomas [27];Biopsy is a preferred diagnostic tool. It can be done either as fine needle aspiration (FNA) or ultrasound-guided or stereotactic-navigated core needle biopsy (CNB). More recent minimally invasive breast biopsy or vacuum-assisted biopsies allow collection of several samples in one insertion instead of several punctures, which minimizes the spread of potentially malignant cells into surrounding tissue. Larger samples of tissue are obtained by classical surgery (probatory incisions or partial excisions or mamaectomy), as is done with tissue from regional lymph nodes. Tissue collected from breast tumour and sentinel lymph nodes is examined microscopically to determine the pathomorphological features and to classify them [29];Histological proof of malignancy and assignment of histopathological phenotype has been a principal diagnostic method for a long time. It is supplemented by analysis of specific tumour cells products or markers to determine a molecular subtype of BC. Common biomarkers currently include oestrogen (ER) [30] and progesterone (PR) receptors [24], cytokeratins (CK) [19,31,32], human epidermal growth factor type 2 receptor (HER2) [18,33,34]. The BC samples obtained by biopsy and/or from post-surgery specimen can be currently processed by various methods (described in the following section). Genomic tests using individual or multigene assays can detect expression patterns of candidate genes associated with BC. All these methods should determine whether cancer is present, and if so, to identify the type of tumour, location, shape and spread of masses within or outside of the breast, respectively [4,10,19,21].

Therapeutic and prognostic factors include tumour size, grade and lymph node infiltration [26]. Tumour size (≤2 cm vs. >2 cm) is a prognostic factor for local or regional recurrence and overall survival and the probability increases with a tumour diameter [26]. Nottingham cancer grading system (score from 1 to 9) scores nuclear pleomorphism, mitotic activity and other features [26]. Lymph node status (positive or negative) refers to a presence or absence of tumour cells in close or distant sentinel nodes. Control of sentinel lymph nodes is a standard procedure (through a surgery or later by PET) to predict a recurrence of disease. We should consider that up to 1/3 of untreated patients with negative lymph node status will develop recurrent or metastatic disease within 10 years after diagnosis [26]. All above data and other clinical evidence of eventual distant organ metastasis are a part of tumour–node–metastasis staging system. BC can be divided into four clinical stages—I, II, III and IV [6]. Cancer staging provides information useful for individual patient prognosis (aggressiveness of cancer and the extent of its invasion) [35] and for large scale analysis [36];

**(c) *Monitoring tests*** during and after treatment are done to determine the benefit of therapy and may be used to check for any signs of recurrence. Examination of blood samples for increased levels of serum biomarkers from tumour cells are commonly used in this step, e.g., carcinoma antigen 15-3 (CA 15.3), Carcinoma antigen 27.29 (CA27.29), Carcinoembryonic antigen (CEA) and others [2]. Periodical NMRI and mammography tests should be used in controlling potential relapse of tumours in their original site, new tumours in other breast or potential spread into distant metastases (Figure 1). Patients using aromatase inhibitors undertake regular densitometry examination of bone density. Regular check-ups are usually scheduled every 3–4 months during the first two years, every 6–8 months during 3–5 years’ period after the treatment, and after that once a year [24,37].

## 3. Histopathological Forms of BC

Determination of histopathological features of tumour mass is a principal step in diagnosis and in determining suitable therapeutic algorithm and prognosis. In many cases, non-invasive procedures such as mammography and MRI can provide enough data to distinguish non-tumour lesions from tumours masses and benign from malignant breast tumours, respectively (Figure 1; red vs. black labelled items) [10,27,28]. Nevertheless, tumour samples obtained by multifocal biopsy, incision, or post-surgery samples (after partial or total ablation of breast) are inevitable for diagnostic conclusion about malignant histological phenotype [29]. Breasts are made up of fatty tissue, fibrous tissue forming the stroma and glandular tissue. Pathogenically, the most important structure for understanding development of BC is terminal ductal-lobular segment (Figure 1) [4,10]. BCs arise from multipotent breast stem cell precursors which give rise to myoepithelial basal cells (allowing milk ejection) and luminal epithelial cells (milk production). These two histological phenotypes—basal and luminal—differ also by their biological functions and expression of specific genes, which also largely determine therapeutic responsiveness of specific tumour cell lineages. Mixed basoluminal type also exist and share certain features of both main types [19]. There are several histopathological subtypes of pre-cancerous and invasive BCs; their main features are as follows: 

**(1) Non-invasive (in-situ) types of breast tumours** remain in a particular location of the breast, without being spread into surrounding breast tissue, lobules or ducts [6]. The two main types of in situ cancers are recognized: ductal carcinoma, which represent 80% of pre-cancerous forms, and lobular carcinoma accounting for the next 20% [39].

(a) Ductal carcinoma in situ (DCIS), also called intraductal carcinoma (or stage 0 BC), represents nearly 20% of newly diagnosed breast tumours altogether. Recurrence is less than 30% within 5 to 10 years after the diagnosis [6]. DCIS is a pre-invasive form of BC (pre-cancer), that may turn into adenocarcinoma. Tumour is derived from the epithelial cells lining the milk duct. Cell mass grows within the ductus but over the time it can break through the ductal walls into the surrounding tissue. DCIS is divided by histology into papillary, solid, micropapillary, cribriform and comedo-like subtypes, respectively. The low-grade DCSI is characterized by small, uniform-looking cells with uniform nuclei and a normal chromatin pattern. The intermediate-grade DCSI is similar to the low-grade, the difference between them being the intraluminal necrosis found in some of the ducts. The high-grade DCSI has atypical pleomorphic cells with very distinct nuclei. Necrosis also occurs and is mostly surrounded by proliferating tumour pleomorphic cells. Lumpectomy without BC radiation therapy has 25–35% chance of recurrence. Adding radiation therapy to the treatment decreases this risk to about 15% [39]. 

(b) Lobular carcinoma in situ (LCIS) grows from adenomatous cells inside milk producing lobules of breast and tends to remain within lobules [6]. As being deeper in the breast it is rarely felt as a lump. It is rather detected randomly in preventive mammogram or in needle or excisional biopsy samples due to other reasons. Histologically, LCIS is described as classic, pleomorphic, and florid (with central necrosis) subtypes, respectively. The latest two show an increasingly higher rate of anaplasias and signal increased risk of malignant transformation (7–12 times higher as compared to classic type) [39]. Another benign lobular pathology is atypical lobular hyperplasia (ALH). Both increase risk for invasive BC.

**(2) Invasive BCs** grow invasively into surrounding breast tissue, spread into lymph nodes and other organs. Two main types of malignancies can be classified as invasive lobular carcinoma (ILC) and invasive ductal carcinoma (IDC). There are also tumours which share the features and cell types of designated cases as mixed forms of carcinomas.

(a) IDC (also called infiltrating ductal carcinoma) is the most common type of invasive breast tumour (accounts for 70-80% of all cases). It occurs mostly in women older than 50 years and mainly those with inherited *BRCA1* and/or *BRCA2* mutations [13]. The risk of IDC is elevated if the first menstrual period came before age 12 or if the woman entered menopause after age 55, due to prolonged exposure to female sex hormones. The 5-year survival rate estimate is 99% when cancer has spread only within the breast or 86% if cancer has spread into neighbouring lymph nodes [2]. Prognosis is getting dramatically worse (falls to 28%) if it has metastasized to distant parts of the body. IDC outgrowths outside of the milk ducts to other parts of the breast making a solid mass that the patient feels as the breast lump. It spreads through the lymph vasculature or bloodstream. It is a heterogeneous group of tumours with several subtypes based on morphological properties of tumour cells [39]: (i)The classical nonspecific subtype is typical by pleomorphic cells with different shapes, sizes, and large non-uniform nuclei. In most cases, squamous and apocrine metaplasias, tissue necrosis and calcification are observed;(ii)The apocrine subtype is associated with a very poor prognosis. Cells are large, with typically strongly acidophilic granular cytoplasm. The nuclei are distinct and vesicular [39];(iii)Medullary carcinoma accounts for 3–5% of BC. Typically, women in their late 40s and early 50s are affected, and most commonly those who carry a *BRCA1* gene mutation. It is often of triple-negative molecular pattern, but more responsive to chemotherapy and better prognosis than other ductal cancer;(iv)Mucinous carcinoma, also called colloid carcinoma, accounts for less than 2% of BC. Tumours contains clusters of uniform epithelial tumour cells with mildly atypical nuclei that are loosely surrounded by excessive mucus;(v)Papillary ductal carcinoma accounts for less than 1% of invasive BC. It is typical for older, postmenopausal women. Under a microscope, these cells resemble tiny fingers (papillae). Cells are typically small;(vi)Tubular ductal carcinoma accounts for less than 2% of BC and is more common in women older than 50. The tumour cells are oval or elongated, well differentiated, randomly arranged, and lined with a single layer of epithelial cells and without the outer layer of myoepithelial cells. In all these last three phenotypes tumour cells are positive for ER and/or PR receptors and negative for the HER2 receptor [24].

(b) ILC is the 2nd most common type of invasive cancer (~10% of cases). ILC starts in lobules, in 1/5 of cases in both breasts, and is harder to detect on mammograms. Cells are small, relatively uniform and of rounded shape, and have a typical stromal infiltration pattern. Like ductal carcinoma, lobular carcinoma can also be divided into subtypes according to closer histological characterization:(i)Classic (non-specific) subtype carries typical morphological features of lobular invasive carcinoma. Cells are small and uniformly distributed across the stroma, forming a typical Indian pattern. All, or at least part, of the pleomorphic subtype cells are considerably larger than those of the classical subtype and are characteristic for their eosinophilic cytoplasm. The nuclei of these cells are hyperchromatic, located eccentrically within the cell and with a very pronounced nucleolus. Absent expression of hormone receptors and high expression of tumour protein p53 and HER-2 receptor are also very typical for this subtype [40];(ii)Tubulolobular subtype is a variant of classical lobular carcinoma. It is characterized by small tubular formations with and without a lumen and cells forming a linear pattern similar to the classical subtype. An in situ lesion is often present in this subtype;(iii)Histiocytoid subtype consists of cells with a diffused growth pattern. Tumour cells are large, with a foamy cytoplasmic consistency that contains a significant number of granules. E-cadherin expression is negative for this subtype [39].

**(3) Special types of breast tumours.** This category contains rare and histologically or clinically distinct BCs: 

(a) Inflammatory BC (IBC) is typical by erythema occupying at least one third of the breast and “peau d’orange-like” changes on the skin. IBC is considered a specific histological subtype, however, with no specific molecular signature. Thus, the diagnosis is based on clinical signs and symptoms [41]. IBC is a very aggressive, fast-growing type of cancer that accounts for 1% to 6% of invasive BCs. It arises from ductal malignant cells and usually has a high histological nuclear grade. As it grows more superficial into the soft tissue of breast it often blocks lymph vessels causing the breast to get inflamed. Skin upon the lesion becomes erythematous, warm, and swollen due to increased blood flow and accumulation of exudate with white blood cells build-up. Surface is getting thicker and firm with dimpled appearance like an orange peel [42]. The breast is getting tender or painful and itchy. Sometimes there are large patches of redness and red bumps that resemble bug bites (“inflammatory BC rash”). Only 15% cases have a real lump. When spread to areolar-nipple segment it causes the nipple to be flattened or inverted. Typically, axillary lymph nodes and subclavian lymph nodes are early getting swollen. The onset of the symptoms is relatively rapid and has to be distinguished from benign bacterial infections or inflammation around the cysts. Hypothesized genes contributing to IBC´s aggressive phenotype includes *ESR1, GATA3, MUC1, ERBB2* and *KRT5* [41];

(b) Paget’s disease of breast nipple is a rare BC occurring mostly around 56 years of age, and accounts for 1% to 4% of all invasive cases in woman. Nevertheless, this kind of BC also occurs in males [43]. Tumour originates from the ductal cells (in situ or invasive ductal carcinoma) and spreads to the skin of areola and the nipple with invasion of the overlying epidermis by malignant cells (so called Paget cells). Clinically the tumour is first manifested by eczematous changes, nipple discharge, bleeding, itching that can be misdiagnosed as psoriasis, contact dermatitis, erosive adenomatosis of the nipple etc. [44]. These finding are later followed by palpable mass of tumour [43]. Prognosis for males is worse, as five-year survival rate in males is 20% to 30% compared to 30% to 40% in females [3,44];

(c) Angiosarcoma of breast is a rare tumour accounting for less than 1% of all BCs, and 1% to 2% of all body sarcomas. It is formed up from the endothelial lining lymph or blood vessels within a breast. Angiosarcoma is most common in people older than 70 and it can occur as a complication from radiation therapy to the breast with some 8 to 10 years delay. It is often diagnosed late when it has already spread to other areas of the body [39];

(d) Phylloides tumour is a rare, mostly benign breast tumour (up to 80%) which mostly affects women in their 40s, though it may develop in patients of all ages. The tumour develops from the cells of the connective tissue (stroma) of the breast. Approximately 20% to 25% of phylloides tumours may turn to malignant phenotype. People with Li–Fraumeni syndrome (AR-inherited condition) are at an increased risk for this type of tumour [6].

**Metastatic carcinoma.** Breast tumour cells spread by lymph and blood. Tissues mostly affected by metastasis include brain, bones, lungs, and liver [6] (Figure 1). BC metastases, similar to other tumours, display certain organotropism, which, in addition to histological origin, is also determined by molecular subtype (Figure 2). BC spreads through lymphatic drainage into closest lymph nodes, mostly to axillar nodes (~30–50%) or mammary internal chain lymph nodes (10–40%), but rarely into supraclavicular nodes (~up to 4%). Peritoneal metastasis incl. ovaria represent near ~10% of all metastases. Most of them originated from lobular carcinoma (40%). If cancer has spread only within the breast, the 5-year survival rate reaches up to 99%. When the neighbouring lymph nodes are affected the survival rate decreases to 86%, and if it has spread to a distant part of the body, the rate decreases to 28% [38]. 

## 4. Molecular Subtypes of BC

In a study showing the differences in gene expression in various BCs, Perou and Sorlie proposed for the first time the “Molecular Classification of BC” in 2000 and brought a molecular insight into histopathological classifications [24,46]. Except for normal breast cell, several sub-groups of cancer cells were proposed, as luminal cancer (express genes typical in normal luminal epithelial cells), *HER2* positive (overexpress ErbB2/neu oncogene) and basal cancer (express genes typical in normal breast basal and myoepitelial cells) [21,23,33]. The classical, mainly immunohistochemical markers analysed from BC cells used in the classification include ER, PR and HER2 (overexpressed oncogenic variant of EGF-like receptor) receptors. Additional markers used mainly for basal cell carcinoma stratification include Ki-67, EGFR (epidermal growth factor receptor) and basal cytokeratins (CK5/6, CK14, C19). Additionally, Urokinase plasminogen activator (uPA) and plasminogen activator inhibitor (PAI) could be analysed [9,18,23,24,30] (Figure 2 and Figure 3). 

Molecular classification strengthened the view on BC as histomorphologically and biologically heterogeneous group of tumours that show different growth intensity, invasiveness, and metastatic imprint, which require different therapeutic strategy [23]. As research progresses, the classification is a subject of continuous updates and modifications. In particular, 2–3 subforms of luminal BCs were established [47] and basal cell types of cancers were reclassified into triple negative (TN) BCs with “basal-like” and “non-basal like” features using EGFR and CK5/6 markers, respectively, and their subtyping continues (Figure 2) [19,48]. Each BC seems to be morphologically and genetically unique, thus determining that the clinical manifestations of the disease are needed for personalized therapy for each patient [49].

ER and PR are intracellular (nuclear) receptors for oestrogens and progesterone in humans, encoded by gene ESR1 (Ch 6q25.1-q25.2) and PGR (Ch11q22), respectively. After hormone binding complex hormone/receptor is formed and assembled into dimers. After being actively transported into the nucleus, dimers serve as transcription factor and bind to hormone responsive elements of responsive genes to activate their transcription. ER and PR are common constituents of hormone-responsive breast tissue cells, mainly of luminal-epithelial origin as compared to basal myoepithelial lineages [50]. Oestrogen signalling is involved in control over cell growth, proliferation, and differentiation of breast tissue. According to the hormonal receptor status and sensitivity to hormone therapy, BCs are designated either as hormone positive (H+), as seen in luminal types of cancers, or as hormone negative (H−), as typical in basal types of cancers and BCs (Figure 2). Hormone-positive BCs could be targeted by hormonal therapy [2,10]. Mutations of ESR were detected in spontaneous BCs [30]. 

HER2 (human EGF-like receptor; ERB-B2) is an oncoprotein-surface receptor with tyrosine kinase activity from EFFR (epidermal growth factor receptor) family encoded by oncogene *neu/ERBB2* (alternatively erbB-2, CD340, HER-2/neu) on locus Ch17q12 [16,34]. HER2-positive BCs overexpress the HER2 gene via amplification (20-25% of BCs) or other ways (~5%) [33]. Amplification is detected by immunohistochemical and hybridisation techniques. There are likely more than 60% of BCs considered yet as HER2-negative that possess at least some HER2 proteins on the surface (so called *HER2-low*) [23,34]. HER2 promotes tumour growth by enhancing cell proliferation, invasion, and metastasis by constitutively activating classical ras-dependent signalling pathway (ras-raf-MAPK) and alternative pathways (e.g., PI3K/AKTPKB). HER2+ is marker of possible responsiveness to targeted monoclonal therapy [23,26,33].

Cytokeratins are components of intermediate filaments of cell cytoskeleton and are inevitable in epithelial cells to resist mechanical stress. Their expression is tissue specific are used to identify the origin of human tumours. Presence of CK5/6, CK14, CK18 and CK19 is one of the diagnostics markers of cancer from basal cells, i.e., triple negative BC [32,48,51,52] (Figure 3).

**Figure 3 cancers-14-05444-f003:**
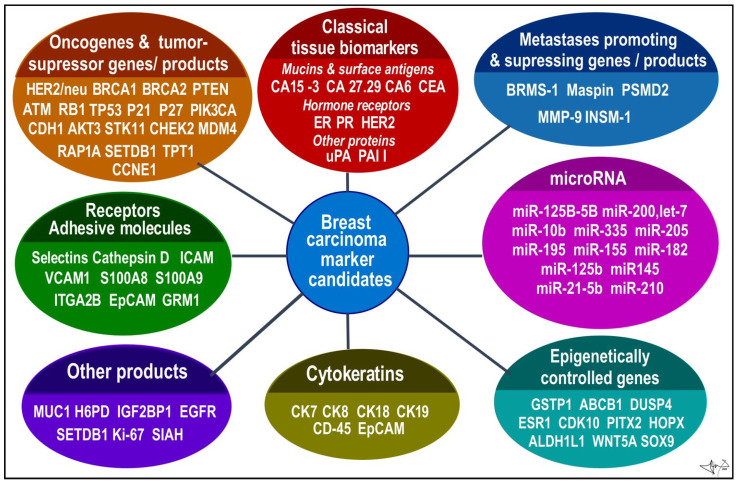
BC candidates arranged according to functional categories. Data were collected from several sources [9,12,13,20,23,24,25,52,53].

*Ki-67* is 359-kD nuclear protein associated with ribosomal RNA synthesis (gene MKI67; Ch10q26.2); it is considered a proliferation index and has been associated with poor prognosis [26]. Overactive MKI67 is identified in most proliferative cells (mainly in S phase), and in opposite is absent in the resting cells (G0). Ki-67 as well as EDGF, which is a natural growth promoting factor, are important in subgroupings especially in ER-positive BCs, although appropriateness of these proliferation markers or more detailed mitotic index scoring system over classical histopathologic predictions is still disputed [21,23]. The uPA is an enzyme that like serum tPA converts plasminogen to plasmin, action that is inhibited by PAI-1. This uPA is an extracellular matrix (ECM)-degrading protease, together with plasmin, in concentrations found in tumour tissue, degrading components of ECM in tumour stoma (fibrin, fibronectin, laminin etc.) and enhancing invasive growth and metastasing. Both PAI-1 and uPA belong to progression and metastasis markers of BC to predict the benefit effectiveness of adjuvant chemotherapy in patients with early BC, and are used as prognostic biomarkers in lymph node-negative BC [23,24,54].

According to expression of above-mentioned markers, BC can be classified into several molecular subtypes, which show distinct biological features and also invasiveness, therapeutical sensitivity and affinity to various tissues for distant metastases (Figure 2) [19]:

(a) Luminal A BC accounts for up to 50% of all invasive BCs. They express high levels of hormone receptors (either ER or PR or both) and obvious luminal (low molecular weight) cytokeratins. They have low status of HER2 (negative) and express low levels of Ki-67 protein. Histologically they are identified as tubular carcinomas, cribriform or classic lobular carcinomas, or low grade invasive ductal carcinomas, respectively. The most descriptive molecular feature of lobular carcinoma is the loss of E-cadherin. Luminal A cancers grow more slowly and are show lower grade of malignity than other BCs. They respond better to hormone therapy and also have a better prognosis [39,55]; 

(b) Luminal B BC accounts for 20–30% of cases. Similar to type A, these tumours express luminal cytokeratins and are ER-positive. They are mostly PR-positive but could be PR-negative, too. Opposite to A-type, Luminal B express high levels of Ki-67 and their HER-2 status is variable; it can be either negative or positive. Luminal type B often manifests genomic instability and the accumulation of *TP53* mutations. Cancers of this type have a higher proliferation rate and histologic grade than luminal A type. Histologically, luminal B include mostly invasive ductal carcinomas or micropapillary carcinomas. Their response to endocrine therapy (tamoxifene, aromatase inhibitors) as well as prognosis are not as good as in Luminal A [39,55]. 

Recently, BCs with mediate-to high-levels of ER and PR-receptors and HER2-positive status were classified as a special subcategory of luminal B type BCs (luminal HER2(+)). These so-called triple-positive cancers (TP) (ER/PR/HER2-positive) can be treated with hormone drugs as well as drugs that target HER2 [56];

(c) HER2-positive BC accounts for 15–20% of invasive BCs. It is characterized by overexpression of the HER2/neu oncogene, obviously with low expression of both ER and PR receptors. As compared to luminal type A, HER2-positive tumours are associated with worse prognosis. Histologically, HER2+ tumours are high grade invasive ductal carcinomas. They show diffused *TP53* mutations, high proliferation and histologic grade and nodal positivity. HER2+ cancers show more aggressive behaviour/higher mitotic activity, increased invasiveness, cellular motility, leading to earlier and more frequent relapses of the disease after primary multimodal treatment [23];

(d) Triple-negative (TN) or basal-like BC represents ~15% of invasive BCs and is characterized by *ER/PR/HER2-negative* profile (triple negative) and high Ki-67 proliferative index. TNBC is more common in younger women and is more frequent in African than Caucasian population. It shows a high proliferation rate, diffused *TP53* mutation, *BRCA1* mutation (germline, sporadic) and is considered more aggressive than either luminal A or luminal B cancer with very poor prognosis [6,39,55,57]. TNBC is not only a most aggressive subtype, but also stands for the most heterogeneous group as well. Lack of ER, PR removes hormonal feedback control overgrowth a differentiation, rendering TNBC high mitotic activity, high staged nuclear grade and unresponsive to hormonal therapy. Lack of HER2+ makes TNBC tolerant of monoclonal targeted therapy.

Over past decades, TNBC attracted a lot of research interests, and stratification of TNBC is continuously updated. Based on markers as EGFR or CK5/6, TNBC can be subtyped into prevalent basal-like type (approx. 80% of cases) and non-basal like type (approx. 20% of cases) [19]. Several additional categories were proposed considering unique histological features and typical genetic abnormalities caused by cumulative mutations of BC-associated genes (e.g., *BRCA1*, *PTEN*, *RB1*, *TP53*, *NF1*, *HRAS*, *MAPK*, *STAT4*, *SMAD4*, *PIK3CA*, etc.). Subclasses of basal TNBC include basal-like 1 subtype (histologically mostly ductal BC) and basal-like 2 subtype (ductal, squamous and inflammatory BC). Non-basal TNBCs contain cells with mesenchymal or stem-cell like features and immune response genes (so called claudin-low, mesenchymal-like, immunomodulatory subtypes), as well as luminal subtype of TNBC (called LAR) with high androgen -receptor (AR) positivity [53].

## 5. Novel Specific Molecular Biomarkers in Current Use and Future Perspective

As demonstrated before, BC is genetically and biologically a heterogeneous family of tumours [19]. Such heterogeneity may apply not only for interindividual differences, but also for intraindividual differences in a tumour itself within different stages of disease as a result of accumulating new mutations [23,24]. Such changes in tumour behaviour can be noted in clinical course and responsiveness to therapy or recurrence of disease [2,6,18,24,58]. The aim, currently, is to search for a wider range of molecules and/or their combinations on the level of DNA, RNA, proteins or even metabolites, which may serve as credible biomarkers in post-diagnostic, therapeutic and maintenance stages of cancer disease [20,21,24]. A suitable molecular biomarker should be stable, sufficiently specific, easily detectable, and its use should be universally applicable to all patients [21,23].

In addition to tumour-specific molecules arranged from cancer tissue samples, recent advances in molecular methodologies, analytic precision, and sensitivity, allow us to look for primary or metastatic tumour “fingerprints” in blood and other body fluids [20,21,25]. Tumour cells often lose many normal tissue determinants, but in opposite produce a myriad of molecules to be employed as biomarkers, e.g., metabolites, circulating cell-free macromolecules (RNA, DNA, proteins), circulating microRNA (miRNAs), circulating carcinoma antigens (CAs) and circulating tumour cells (CTCs). They can supplement traditional histological methods for diagnosis, prognosis, and tumour recurrence detection on a molecular level [21,59]. These biomarkers can be determined from peripheral blood (minimally invasive) or from non-invasively obtained body fluids such as tears, breath, nipple aspirate fluids, apocrine sweat, and urine [9,20,25] (Figure 4).

(a) RNA

Dysregulated non-coding mRNAs, such as miRNAs (micro RNA), lncRNAs (long non-coding RNA) and circRNAs (circular RNA) are studied as possible markers of multiple cancers. 

The miRNAs are small linear non-coding single-stranded RNA molecules (19 to 24 nt). Through complementary binding they regulate gene activity in cells and affect a whole spectrum of molecular processes at the subcellular level. These miRNAs are studied in the context of cancer [61,62] due to their involvement in the regulation of proliferation, differentiation, migration and apoptosis. The number of miRNAs differs in healthy and malignant cells; therefore, different types of tumours can be characterized by an altered representation of miRNAs [63]. Recent studies have pointed towards dysregulation of specific onco-miRNAs associated with BC–*miR–10b*, *miR–125b*, *miR145*, *miR–21*, *miR–155, mir–30*, *let–7*, *miR*–*25–3p* [64,65,66]. *miR-130a*, *miR-90b*, *miR200b*, and *miR-452* have been shown to regulate drug-related cellular pathways and thus enable tumour chemoresistance. The *miR-221* and *miR-222* are specifically connected with chemoresistance to fulvestrant, doxorubicin, or trastuzumab and *miR-320a* chemoresistance to paclitaxel [67]. For miRNAs to be routinely used in BC prognosis, it is necessary to determine stage-specific miRNAs dysregulation and to detect a panel of single miRNAs for every particular stage [10]. Recent studies have also shown an epigenetic effect of miRNA on ER expression in BC through direct and indirect mechanisms. Examples would be miRNA-142-3p and MiRNA-148a, where miRNA-142-3p down-regulates ER expression by directly binding to the 3´UTR region of ESR1 messenger RNA and miRNA-148a regulates ER expression indirectly through targeting of DNMT1, resulting in upregulation of ER [61].

The lncRNAs are linear non-coding single stranded RNA molecules with a length of more than 200 nucleotides, involved in regulation of transcription, translation, and cell cycle. In tumour cells, lncRNAs are associated with acquiring invasive properties, metastasis, and resistance to chemotherapy. Few lncRNAs have been particularly associated with BC. H19 is an oestrogen induced lncRNAs that has been shown to have a role in cell survival and proliferation. *RHPN1-AS1* (RHPN1 antisense RNA 1), induced by *KDM5B*, has been shown to promote BC via RHPN1-AS1/miR-6884-5p/ANXA11 pathway [20]. 

The circRNAs are non-linear non-coding single stranded RNA molecules in a form of covalently closed continuous loop. Dysregulation of these molecules has also been associated with BC. Out of 1155 differentially expressed circRNAs in BC, 715 were overexpressed and 440 were downregulated when compared with healthy tissue samples. These circRNAs can regulate gene expression via microRNA sponging, therefore their dysregulation can increase proliferation, invasiveness, and migration of tumour cells. Namely *hsa_circ_0072309* [68] and *hsa_circRNA_0001785* [69] have recently been introduced as promising prognostic biomarkers in BC. 

The detection of non-coding RNAs is quite problematic, since their content in body fluids is not large enough to detect, but with *RT-qPCR* the amount of expression of selected non-coding and also non-coding RNAs in the sample can be detected in real time [20]. Besides non-coding RNAs, RT-qPCR can be used to detect cancer-specific methylation patterns and point out genes responsible for cell cycle regulation, DNA repair, adhesion, and metastasis with an altered methylation. Monitoring methylation changes appears to be important in the context of a patient’s response to treatment. *BRCA1* hypermethylation could predict tumour susceptibility to treatment with PARP inhibitors. Hypermethylation of the *GSTP1*, *ABCB1* and *DUSP4* genes could indicate sensitivity to doxorubicin treatment and hypermethylation of the *ESR1*, *CDK10* and *PITX2* genes resistance to oestrogen inhibitors [70].

(b) DNA

Fragments of tumour genomic DNA, also called circulating tumour DNA (ctDNA), contain the same characteristic gene mutations as a primary solid tumour, therefore ctDNA in peripheral blood has a potential to become BC prognosis biomarker and can be used for identifying relatively early-stage tumours [9,71]. Amount of ctDNA in peripheral blood is relatively small, therefore qualitative, and quantitative methods of ctDNA detection are based on NGS and RT-qPCR. Elevated levels of ctDNA have been linked with advanced-stage BC and metastasis as revealed in a longitudinal study of 21 patients. A measure of the ctDNA percentage in samples, a molecular tumour burden index (mTBI), has been positively associated with tumour size and disease progression [72].

NGS can be used for genomic profiling of tumour tissue to reveal intertumour heterogeneity between patients diagnosed with early-stage BC and among patients experiencing relapse [60]. Mutations in certain genes were subsequently associated with specific molecular subtypes of tumours. Somatic mutations of *TP53, PIK3R1* and *NF1* genes were more detected in patients with triple negative tumours. Increased copies of *CCND1*, *FGF3* and *FGFR1* genes were associated with patients with luminal tumour type [40]. 

A recent study used NGS for genomic profiling of metastatic BC patients. Mutations contributing to tumour malignancy were observed in 74% of the participating patients, and in 43% of these patients, changes in the previous treatment were proposed (and in some cases carried out). Their results pointed to the association of tumour metastasis with mutations in specific genes, of which the PIK3CA gene was the most frequently mutated, present in 52% of patients [73].

DNA in extracted tumour tissue can be used for the determination of HER2 status by in situ hybridization with probes labelled fluorescent, chromogenic or silver dyes. Labelled probes are applied to the tissue and bind to the complement copies of the *ERBB2* gene in the cell nucleus. Confirmation of the positive HER2 status allows physicians to recommend treatment against HER2 only to patients who will benefit from it [74].

(c) Proteins

Protein biomarkers such as hormone receptors, basal cytokeratins and other serum markers can be monitored by enzyme-linked immunosorbent assay (ELISA), mass spectrometry and immunohistochemistry in order to compare the physiological and pathological states. Immunohistochemistry (IHC) detects the over-translation of HER2 proteins and other protein on the surface of tumour cells associated with BC prognosis and treatment [75]. IHC can also determine whether the tumour is benign or malignant and ductal or lobular. Through IHC, an interstitial infiltration status can be assessed [20]. 

A novel and promising protein biomarker is serum apolipoprotein C-I (apoC-I), through which a TNBC tumour can be distinguished from non-TNBC tumour since the expression of apoC-I mRNA and proteins is upregulated in TNBC tumours [60]. 

Recently, assays monitoring a combination of more serum proteins have proven to be more diagnostically useful. A 4-test combination of serum proteins CEA, CA125, CA15-3 and TAP (transporter associated with antigen processing) has shown a high sensitivity and can potentially be used as an auxiliary test in an early screening. Assay determining expression of RAD50, PARD3, SPP1, NY-BR-62, and NY-CO-58 antigens could distinguish BC patients from healthy controls. Tumours with *BRCA1* mutation can be separated from tumours with sporadic BC mutations and cancer-free patients with *BRCA*1 mutation via evaluation of KNG1_K438-R457_ and C3f_S1304-R1320_ peptides [75].

(d) Lipids

Lipid molecules are involved in several physiological cellular processes and have a variety of different functions in cells including storage of energy and acting as a cellular signalling molecules, and they are an important structural component of cell membranes [76]. Specific alterations in extracellular lipid levels have been associated with breast cancer. Recent study has shown that low levels of high-density lipoprotein cholesterol (HDL cholesterole) have been associated with higher risk of TNBC, whereas higher levels of low-density lipoprotein cholesterol (LDL cholesterole) have been associated with a higher risk of luminal type B tumours. HER-2 subtype has been associated with higher levels of triglycerides (TAG) in postmenopausal women [77]. Cancer cells can use energy generated from oxidation of fatty acids, therefore extracellular lipids serve as a significant driving motor in the progression of breast cancer. Dysfunctionality of adipose tissue caused by obesity can result in altered levels of inflammatory cytokines, growth factors (insulin, insulin-like growth factor 1), steroid sex hormones (mainly estrogen) and adipokines (leptin, adiponectin, visfatin) which can further stimulate tumour growth [22,78]. A 27-hydroxycholesterol, endogenous selective estrogen receptor modulator, is an experimental lipid BC biomarker, providing a new perspective on BC prevention strategies [75]. 

(e) CTCs

CTCs are rare cancer cells (one CTC per billion normal blood cells) analysed from blood and are informative of tumour progression. CTCs enter the bloodstream through active intravasation or passive shedding from tumours and their presence in early stage increases the risk of tumour recurrence [10]. CTCs can be detected and analysed by immunomagnetic separation and immunofluorescence/enzyme-linked immunosorbent assays (based on their physical properties) and by RT-PCR. CellSearch^®^ (Janssen Diagnostics) is semiautomated antibody-based assay based on immunofluorescence and flow cytometry. Analysis of CTCs using CellSearch^®^ is semiautomated and based on immunofluorescence and flow cytometry. During analysis, the CTCs are enriched by EpCAM (epithelial cell adhesion molecule) antibodies. The CTCs are detected by cytokeratin positivity, positive nuclear staining and CD45 negativity. The other currently greatly studied assay is The AdnaTest^®^ (AdnaGen), which is a RT-PCR based test used for detecting CTCs via GA733-2, MUC1, and HER2 and other mRNA markers specific for cancer. During analysis, CTCs are immunomagnetically enriched with MUC1 and EpCAM antibodies [9]. Because of their rare character and low sensitivity and reproducibility, the CTCs are not recommended solely for BC diagnosis, but they can help differentiate patients with invasive BC from patients with benign tumours [20]. 

(f) Multi-parametric gene expression assays 

In addition to individual gene testing possibilities, there are commercially available assays analysing the expression of selected genes from a large group of candidate cancer genes libraries (more than 250) [20,26]. These tests are designed to characterize the tumour and serve for prediction and/or estimation of growth potency, invasiveness or to estimate the recurrence after treatment and reappearance of tumour in original or other locations. The advantage of these tests is fast delivery of results, but the big disadvantage is the fact that they are intended only for women with early-stage cancer [21,26] (Figure 5). 

**The Oncotype DX Breast Recurrence Score Test^®^** (Exact Sciences Corp.) is a 21-gene expression assay (16 BC–related genes and 5 reference genes) suitable for early-stages (stage I, stage II, or stage IIIa), ER(+)/HER2 (−), lymph node-positive (N(+)) or lymph node-negative (N(−)) invasive BCs [79]. The assay employs RNA extracted from formalin-fixed, paraffin-embedded tissue and uses reverse-transcriptase polymerase chain reaction (RT-PCR). It provides a recurrence score (from 0 to 100) within the breast or in tissues outside the breast (distant recurrence), to figure out whether the benefits from post-surgery chemotherapy will outweigh the risks of side effects (obviously hormonal therapy such as an aromatase inhibitor or tamoxifen). In older women (> 50 years old) a score of 0–25 denotes a BC with a low risk of recurrence and the benefits of chemotherapy over risks are few. Opposite, a score of 26 to 100 means that BC has a high risk of recurrence and therapy is recommended despite the risks of side effects. In younger women (< 50 y of age) benefits of chemotherapy are likely greater than the risks of side effects when the score reads 21 and more [21].

**The Oncotype DX Breast DCIS Score Test^®^** (Exact Sciences Corp.) is intended for patients diagnosed with DCIS (ductal carcinoma in situ). Test analyses the activity of 12 genes and calculates a risk of DCIS recurrence and/or the risk of malignant turn into ductal carcinoma in the same breast. DCIS is the most common non-invasive breast tumour pre-cancer. Usually, it is treated by surgical excision and post-surgery hormonal therapy (most cases are hormone-receptor-positive) and/or eventually radiation therapy. Score (from 0 to 100) may also benefit from radiation therapy after DCIS surgery. Score greater than 54 indicates a high risk of recurrence, and the high benefit of radiation therapy [21].

**The BC Index^®^ (BCI) (Biotheranostics, Inc.)** is a genomic test that analyses the activity of 11 genes in early stages (stages I–III) of ER/PR-positive BCs with no involvement of lymph nodes (N-negative disease) or the lymph-node positive tumours N(+) (tumour cells found in 1–3 lymph nodes). Test is prognostic and estimates a probability (%) of BC recurrence in next 5 to10 years (late recurrence), as well as if a patient will benefit from an additional 5 years of hormonal therapy (“yes” or “no”) [26,80].

**The EndoPredict test^®^ (Myriad Genetics, Inc.)** is used to predict the risk of distant recurrence of early stages (stage I-II), ER/PR-positive, HER2-negative BCs that is either node-negative or has up to three positive lymph nodes [81].

**The Prosigna BC Prognostic Gene Signature Assay^®^** (formerly PAM50 test; Veracyte, Inc.) is a genomic test that analyses the activity of 50 genes (so called PAM50 gene signature). It is suitable for early-stage (stages I-II), ER/PR-positive BC. Test may eventually be widely used to help make treatment decisions based on the risk of distant recurrence for postmenopausal women within 10 years of diagnosis of lymph-node negative or positive cancers (1–3 positive lymph nodes after 5 years of hormonal therapy) [21].

**Mammaprint^®^** (Agendia, Inc.) test is monitoring the expression of 70 different most important genes responsible for BC recurrence within 10 years of successful treatment. The test is intended for women diagnosed with early-stage (stage I or stage II; <5 cm in size) invasive or non-invasive BCs which could be either HR-positive or -negative (ER/PR (+) or ER/PR (−)) and lymph node-negative (N(−)) or lymph node-positive (N(+)) cancers (positivity found in three or fewer lymph nodes). Test calculates a BC recurrence score (low risk or high risk). It allows fast analysis to distinguish between various luminal subtypes of BC. The patient’s microarray result is compared with a standard profile with a good prognosis [21]. Women can undertake it regardless of age, and one of the great advantages of the test is the speed (6 days) with which the results are provided [55].

**BluePrint^®^** (Agendia, Inc.) is an 80-gene array test used to analyse early stages of invasive BCs (stages I–II and operable tumours of III stage), which are hormone-positive (ER/PR+) and HER2-negative. Test allows accurate classification of BC into basal (TN–type), luminal, or HER2 molecular subtypes, respectively, to choose the most appropriate personalized treatment. Test also reveals valuable information about long-term prognosis and response to systemic therapy [82].

(g) Gene polymorphism

In addition to germinative and somatic mutations in the aforementioned genes, increased risk of BC, progressive growth, responsiveness to therapy and overall survival rate have been demonstrated to be also associated with polymorphisms (variants) in many candidate genes [83]. These genes are involved in many different processes suggesting heterogenous background of BC pathogenesis, e.g., in resistance to hypoxia (*HIF1α*), immunity (Il-1β, IL 10), metabolism (*CYP1A1*, *CYP2D6*, *CYP2C19* and *CYP17*, *CYP27B1*-genes for the cytochrome P450; MTHFR (folic acid rec.), VDR (vitamin D rec.)), lipid regulation (fat mass and obesity-associated (FTO), adiponectin), antioxidant defence (GSTM1, GSTT1; GSTP1-glutathion transferases)), COMT (catechol methyltransferase), cell cycle (CDKN2A/2B), signalling paths (ESR1, oestrogen receptor, Hedgehog-GLI, PRKAA1, mTOR, OPRM1 (u-opioid receptor) FGFR4, EGF (growth factor receptors, etc.) and many others. Polymorphisms associated with a higher rate of BC were also identified in miRNA and lncRNAs [84,85]. Importantly, increased risk for BC is associated with SNPs (single nucleotide polymorphisms) in virtually all DNA repair mechanisms after genotoxic damage [86]. DNA mismatch repair (MMR) corrects replication and recombination errors and its malfunction in BC is commonly linked to frame-shift mutations in *hMSH6* gene (human MutS homologue; Ch2p16.3) and SNP Gly322Asp in *hMSH2* gene (Ch2p21). Base excision repair (BER) is involved in single-strand DNA break repair after oxidative DNA damage. Many studies confirmed association between BC and SNPs (Arg399Gln) in key gene *XRCC1* (X-Ray Repair Cross Complementing 1; Ch19q13.31) which encodes scaffold protein completing a repair complex for base excision [87]. Nucleotide excision repair (NER) pathway is involved in DNA excisions after radiation or chemical genotoxic effects including those of excessive oestrogen actions in BC. The Lys751Gln polymorphism in one of the key genes *ERCC2* (*alt. XPD*; Ch19q13.32) is one of the most widely studied genetic cancer markers including BC [86]. Homologous recombination repair (HRR) is involved in the repair of DNA-protein cross-links with key components being *XRCC2, CHEK2, ATM, PALB2, FANCA, BRCA1, BRCA2 and RAD51* gene. In addition to well-known BC- linked germinative mutations in *BRCA1, BRCA2, ATM* and *PALB* gene, certain polymorphisms in *BRCA1, BRCA2 and CHECK2* genes, respectively, influence the survival rate in BC [88]. The increased risk for BC is also associated with SNP in *XRCC2* gene (Ch7q36.1), *XRCC3* (14q32.33) and *RAD51* gene (Ch5q15.1) [86]. Higher risk of BC is also associated with SNPs in tumour suppressor genes from *apoptotic machinery* as *p53*, *MDM2* and *MDMX*, *CASP8, PTEN* etc. Although, very recent meta-analysis does not support a previous role of certain SNPs in *TP53* gene (IVS3 16 bp and IVS6+62A) in overall BC- risk [89].

## 6. Diagnostic and Therapeutical Nanotools

Use of nanoparticles (NPs) as diagnostic markers and therapeutic tools in oncology, including breast cancer (BC), became an increasingly attractive topic and the research is advancing rapidly. During their progressive growth, solid tumours including BC, undergo inevitable structural changes including angiogenesis to gain access to nutrients. As with many other features of tumour mutagenesis, neovascularisation is massive but chaotic, erratic, accompanied by structural defects in vessel walls permitting nanocarriers (<100 nm) to extravasate into tumour interstitium though leaky tight junctions [90]. As an advantage, this abnormal tumour-selective endothelial permeability allows gradual accumulation of NPs or NPs-treated drugs in cancer tissue that is further facilitated by defective lymphathetic drainage in the tumours and lower-rate of NPs-phagocytic removal as opposite to the obvious size of particles (>100 nm in diameter). While passive targeting uses the above inherent physical properties, active nanocarriers further improve tumour-lineage specific targeting by using NPs with antibodies or ligands directed towards specific tumour antigens, e.g., anti-HER2 antibody-PEG liposomes targeting HER+ BC [91]. Inorganic, mostly metal based NPs with silver (Ag), platinum (Pt), gold (Au), Au–Ag alloy, Au–gadolinium (Gd) cores appeared to suitable contrast agents (CAs) that provide better signal-difference-to-noise-ratio over classical CAs (e.g., iodine based) when analysing BCs in naturally dense breast tissue using mammographic imaging techniques such as computed tomography (CT) (e.g., Ag Au NPs, near-infrared fluorescence (NIRF) (e.g., Au and AgNPs) or contrast enhanced dual-energy mammography (DEM) (e.g., AuNPs) [92]. Metal NPs can also suppress tumour growth or completely eradicate the tumour tissue locally without systemic side effects. Some NPs (e.g., iron oxide NPs, GdO_2_ NPs, Au-coated Fe_3_O_4_ NPs) show magnetic properties when used in magnetic resonance imaging (MRI). Hyaluronan-modified magnetic nanoclusters (HA-MNCs) as active NPs can allocate CD44 in highly proliferative BC cells [92]. There are also limitations in their use as: 

(a) cytotoxicity (e.g., NPs with Ag, Au, TiO_2_ or GdC_2_ induce excessive reactive oxygen species (ROS) in healthy cells);

(b) natural immunity response (e.g., Au and Au–Ag NPs, carbon nanotubes, GdC2-NPs are phagocyted as antigens to build up immune response over time);

(c) retention in mammary tissue (misinterpret results of control scans) and retention in other organs [90]. 

Organic NPs include liposomes (LIP), micelles, solid lipid nanoparticles (SLN), dendrimers, and protein NP. Due to mutilayered structure these NPs are optimal for drug delivery into BC tissue. Doxorubicin-loaded PEGylated LIP were approved in the mid 1990s, later followed by albumin-bound paclitaxel NPs, and more recently polyglutamate/polyaspartate paclitaxel NPs were tested. Current progress in the research of NPs opens new horizons for even more advanced and targeted diagnostics and safer individualized therapy of breast cancer [93].

## 7. Conclusions

The incidence of BC is constantly increasing and although the quality of healthcare is rising, 5–6% of women diagnosed with BC are in an advanced metastatic stage. The use of molecular methodologies could complement and enrich classical diagnostic procedures. Their undeniable advantages are their inexpensive cost, ease of processing tumour or body fluid samples, and speed of delivery of results. Tracking atypical amounts or types of molecular biomarkers allows for earlier diagnosis and more personalized therapy. MicroRNA dysregulation, methylation profile. New knowledge in this field of research is constantly emerging and represents the future of cancer diagnosis and therapy.

## Figures and Tables

**Figure 1 cancers-14-05444-f001:**
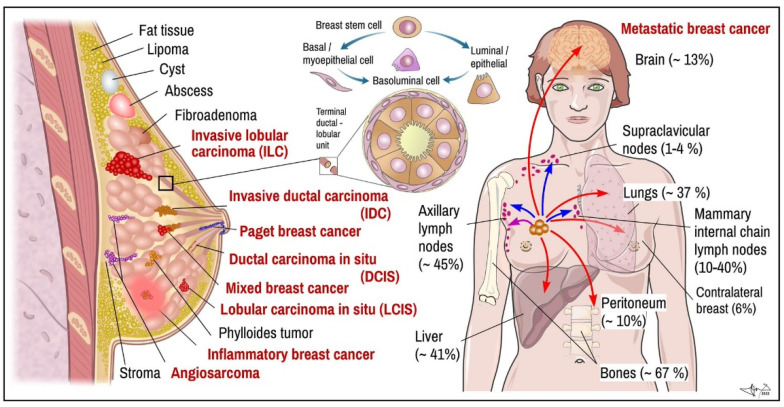
BC. Left: main histopathological cancer types (red) together with other pathological findings. Middle: scheme of terminal ductal-lobular unit illustrating location of basal-myoepithelial cells (milk ejection) and luminal cells (milk production). Right: main targets of BC metastasis (frequency). Arrows indicate metastatic spread: violet-local, blue—via lymph, red—through blood. Data combined [19,21,26,38]. Certain items on right credited to Servier Medical Arts (CC-BY-3.0 licence; https://creativecommons.org/licenses/by/3.0/, accessed on 12 October 2022).

**Figure 2 cancers-14-05444-f002:**
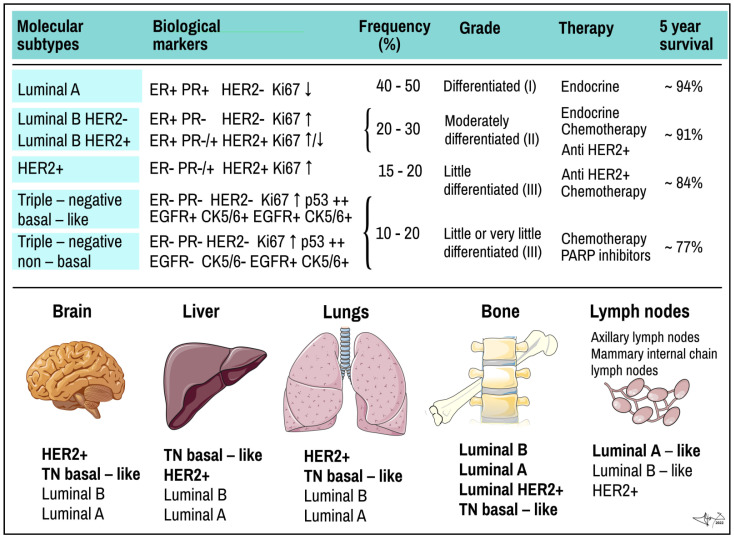
Molecular subtypes of BC and preferred places they metastasize. Upper panel: molecular subtypes of BC. TN, triple negative tumour. ER and PR, oestrogen and progesterone receptor; EGF, epidermal growth factor receptor; HER2+, HER2+/neu/EGF-like growth factor receptor; Ki67, proliferation related antigen; p53, product of *TP53* tumour suppressor gene. Lower panel: most common organ and tissue sites of metastases of different types of tumours. Individual types are arranged in the same order as often they metastasize (in bold the rate exceeds 40%). Data combined [6,9,19,20,25,45]. Some picture items credited to Servier Medical Arts (CC-BY-3.0 licence; https://creativecommons.org/licenses/by/3.0/, accessed on 12 October 2022).

**Figure 4 cancers-14-05444-f004:**
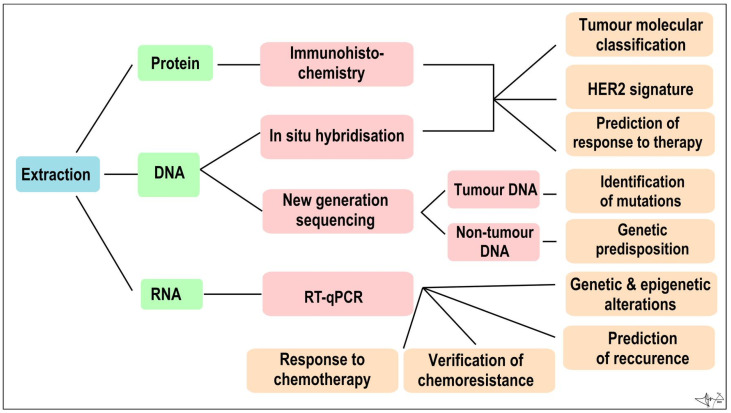
Algorithm of immunohistochemical and molecular genetic analysis of a tumour sample Combined [60].

**Figure 5 cancers-14-05444-f005:**
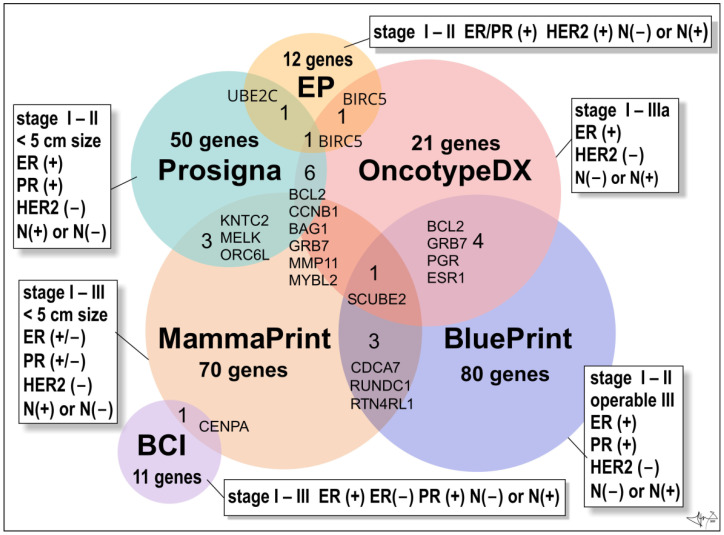
Multigene BC tests. Venn diagram is presenting numbers of genes/proteins evaluated by commercially available multigene tests. Abbreviations: ER(+) and ER(−) oestrogen receptor positive or negative cancers, respectively; PR(+) and PR(−), progesterone receptor positive or negative cancers; HER2(+) overexpression of HER2 oncoprotein; N(−) not present in lymph nodes, N(+) present in 1 to 3 lymph nodes [21].

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
