# Peer review of "Classic and New Markers in Diagnostics and Classification of Breast Cancer"

_cancers, 2022, doi:10.3390/cancers14215444_

Round 1
Reviewer 1 Report
The manuscript “Classic and New Markers in Diagnostics and Classification of Breast Cancer” provide an overview of current knowledge in breast cancer. The authors accurately describe subtypes, molecular and morphological characteristics and the latest aspects in diagnosis and therapy. The bibliography is up-to-date and such an in-depth study in the field in the literature is not yet known.
Authors are advised to make few corrections before the manuscript can be accepted for publication:
Why do authors use italics for some words? E.g. histopathological phenotype (line 112)
Paragraph 2b (diagnostic test) needs to be slightly reorganized. It is very rich in content and difficult to read
Line 519: Are the authors referring to Next Generation Sequencing? Otherwise it would be good to explain what is meant by “New Generation Sequencing”
Minor details:
- Some words that had been abbreviated with an acronym are then put in full along the text (e.g. breast cancer)
- CA 15.3 (Carcinoma antigen 15-3), CA27.29 (Carcinoma antigen 27.29), CEA 140 (Carcinoembryonic antigen) line 140 it would be more appropriate to put the acronym in brackets and the full name before. This should be done along the text.
- Line 281: do the authors mean "endothelial lining"?
Author Response
Ad1) We have used italics as a form of distinguishment, but for the reviewerś reguest we decided to reorganize the manuscript, didn´t use italics and made the manuscript more easy to read.
Ad2) Paragraph 2b has been also reorganized to be more easy to read.
Ad3) Line 519: we indeed were referring to the next generation sequencing, this has been corrected.
Minor details:
Ad1) All words that have been previously abbreviated are then only referred as their acronyms.
Ad2) In line 140, the acronyms have been in put in brackets and the full name has been put before the brackets.
Ad3) In line 281, the endothelia lining has been corrected to the endothelial lining.
Reviewer 2 Report
The review describes classical methods of breast cancer diagnosis, current and emerging, modern molecular methodologies. The paper is well designed and logically written. However, the overall abstract remains quite vacuous and does not list the exact targets. For instance, the old markers ( estrogen receptors, EGFR etc) were not mentioned. The new markers should be listed and focused on as follows : microRNAs ( - which ones are the most significant ones and currently used in the advanced diagnostics?), biomarkers in patients’ body fluids ( which biomarkers?) etc. Authors indicated “… the presence of atypical biomarkers “ – which ones? they were not listed/mentioned (“… expression or methylation profile of selected genes … - which genes/ they were not listed. The abstract should be revised and made more specific.
Section with microRNAs should be extended. I suggest making a new subsection for miRs/epigenetic regulation. Check this paper for references https://pubmed.ncbi.nlm.nih.gov/34831189/
Gene polymorphism ( of important drug metabolizing enzymes) was not discussed.
Lipid biomarkers were not described. The role of lipid metabolism and obesity-related markers are missing too. Check this paper for references https://pubmed.ncbi.nlm.nih.gov/35814391/
Diagnostic procedures with nanotechnology were not addressed /presented properly. This new perspective of diagnostics using nanoparticles and artificial intelligence should be extended as one of the most promising.
Author Response
We would like to express our thanks to reviewer 2 for valuable suggestions. We corrected mistakes and added most of suggested topics in abbreviated way. Considering ample data for every of suggested topics (like epigenetics , miRNA, polymorphisms or nanoparticles) every may be worth a separated review.Round 2
Reviewer 2 Report
I am satisfied with the revised version of this manuscript. Authors addressed all my comments properly.